# A new electrode design for ambipolar injection in organic semiconductors

Thangavel Kanagasekaran[1], Hidekazu Shimotani[2], Ryota Shimizu[1], Taro Hitosugi [1] & Katsumi Tanigaki [1,2]

Organic semiconductors have attracted much attention for low-cost, flexible and human-friendly optoelectronics. However, achieving high electron-injection efficiency is difficult from air-stable electrodes and cannot be equivalent to that of holes. Here, we present a novel concept of electrode composed of a bilayer of tetratetracontane (TTC) and polycrystalline organic semiconductors (pc-OSC) covered by a metal layer. Field-effect transistors of single-crystal organic semiconductors with the new electrodes of M/pc-OSC/TTC (M: Ca or Au) show both highly efficient electron and hole injection. Contact resistance for electron injection from Au/pc-OSC/TTC and hole injection from Ca/pc-OSC/TTC are comparable to those for electron injection from Ca and hole injection from Au, respectively. Furthermore, the highest field-effect mobilities of holes (22 cm$^2$ V$^{-1}$ s$^{-1}$) and electrons (5.0 cm$^2$ V$^{-1}$ s$^{-1}$) are observed in rubrene among field-effect transistors with electrodes so far proposed by employing Ca/pc-OSC/TTC and Au/pc-OSC/TTC electrodes for electron and hole injection, respectively.

[1] WPI-Advanced Institute for Materials Research (WPI-AIMR), Tohoku University, 2-1-1 Katahira, Aoba-ku, Sendai 980-8577, Japan. [2] Department of Physics, Tohoku University, 6-3 Aramaki, aza-Aoba, Aoba-ku, Sendai 980-8578, Japan. Correspondence and requests for materials should be addressed to T.K. (email: thangavel.kanagasekaran.d6@tohoku.ac.jp) or to H.S. (email: shimotani@m.tohoku.ac.jp) or to K.T. (email: tanigaki@m.tohoku.ac.jp)

Carrier injection from metal electrodes to semiconductors and electrical transport in semiconducting active layers are important two key actions in electronic devices. Generally, two types of semiconductors can be considered: One being organic semiconductors (OSCs) based on carbon based molecules and the other being inorganic semiconductors (ISCs), such as silicon and germanium. The former has a clean surface without dangling bonds in a van der Waals bonded system[1–5], and the latter is a covalently bonded system and frequently has dangling bonds on its surface. By reflecting these differences between the two types of semiconductors from their bonding nature, the Schottky limit of metal-semiconductor (MS) interfaces is commonly observed for OSCs, while the Bardeen limit is more often seen for ISCs[6, 7]. Consequently in OSCs, the difference in energy level between the Fermi level ($E_F$) of electrode and the valence band maximum (VBM) or the conduction band minimum (CBM) becomes a Schottky barrier height against hole and electron injection, respectively. $E_F$ of Au is generally located near VBM while that of Ca is near CBM of OSCs, and hitherto hole injection has been more popularly observed when Au electrode is employed, but efficient electron injection needs air-unstable metals with high $E_F$, such as Ca[8–16]. Therefore, it has been one of the major problems for long years that injection of electrons has been greatly difficult compared to those of holes in OSCs, because stable metals with low $E_F$, such as Au, Cu, and Ag, are typically employed. Contrarily, a barrier height for carrier injection is nearly independent of $E_F$ for ISCs owing to the disorder-induced gap states (DIGS) and metal-induced gap states (MIGS) inside a band gap made by disorder/dangling bonds on the surface and MS interface structure[6, 7]. Such a phenomenon is known as Fermi level pinning leading to a vacuum level shift at the MS interface.

Simultaneous injection of holes and electrons (ambipolar injection) is possible in field-effect transistors (FETs) of OSCs. Recombination of holes and electrons generate excitons, and light can be emitted with energy release in a radiative process. Such

ambipolar injection and subsequent light emission have attracted much attention in terms of optoelectronics, because no chemically doped pn junction is required for light-emitting FETs (LE-FETs)[5, 8, 16–21]. Instead, a pn junction, a fundamental component of electronic devices, is automatically made at the recombination zone of holes and electrons in the single-crystalline OSC (sc-OSC) active layer. In particular, LE-FETs of sc-OSCs have created growing interest because of their high mobility. However, serious problems are still left for optoelectronic applications that hetero electrodes (combination of two different metals with high and low $E_F$) are required for highly efficient ambipolar injection[20–26]. Chemical doping to bulk OSCs is reported as a possible method to control the transporting charge polarity, but the chemical doping is not for enhancing carrier injection but for increasing channel transport current by lowering the channel resistance (not for contact resistance), and therefore strongly asymmetric carrier injection between holes and electrons still gives serious problems[27].

Here, we report a new concept of electrode for sc-OSCs, which consists of a bilayer made of a polycrystalline thin film of OSC (pc-OSC), which is made of the same material as a sc-OSC used as an active layer, and linear-chain alkane tetratetracontane (TTC) coated by a metal (M: Au, Ca) thin film: M/pc-OSC/TTC. According to the unique carrier injection mechanism, highly efficient (higher than those of pure Au and Ca) and comparable hole and electron injection is realized. Air stable Au/pc-OSC/TTC electrodes show high ambipolar injection efficiency and bright light emission with very high current density of 25 kA cm$^{-2}$ is obtained.

## Results

### A new concept of electrodes.
An important concept of electrode for carrier injection (Fig. 1b), which is very different from the conventional metal electrodes (Fig. 1a), is proposed. FETs

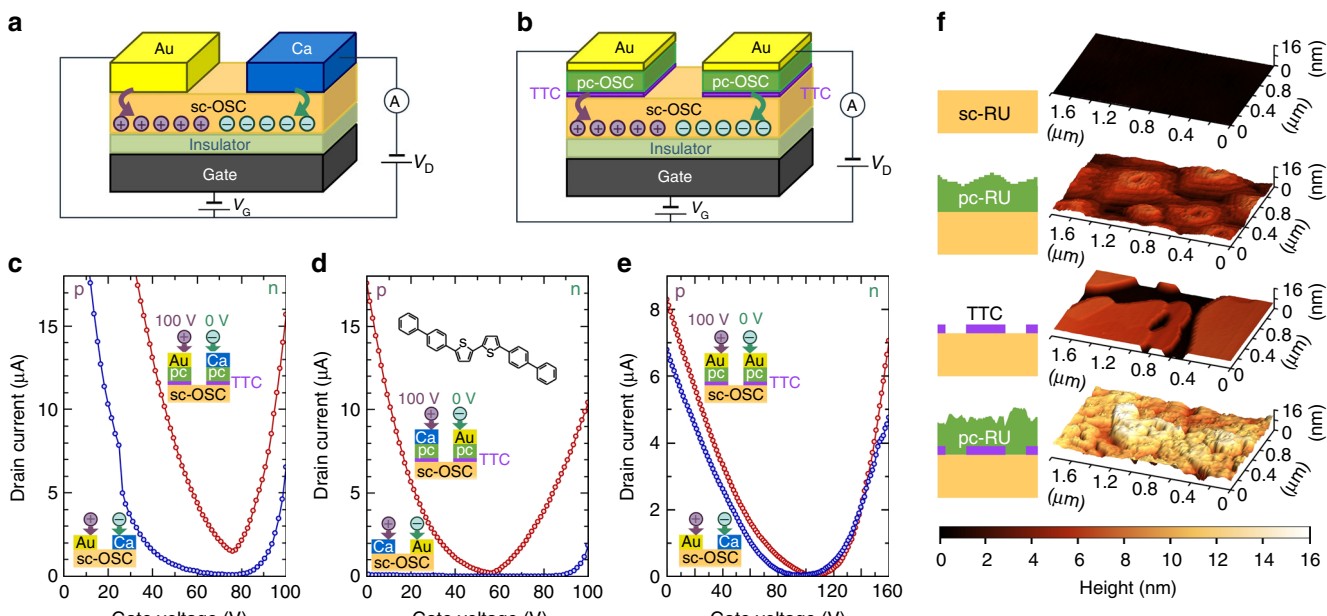

**Fig. 1** Carrier injection performance of conventional and novel electrodes. **a** A field-effect transistor (FET) with Au and Ca electrodes for holes and electrons injection, respectively. **b** A FET with the new electrodes Au/pc-OSC/TTC both for electrons and holes. Carrier injection performance for sc-BP2T FET with hetero electrodes; **c** Au for holes and Ca for electrons (blue circles) vs. Au/pc-BP2T/TTC for holes and Ca/pc-BP2T/TTC for electrons (red circles), **d** Au for electrons and Ca for holes (blue circles) vs. Au/pc-BP2T/TTC for electrons and Ca/pc-BP2T/TTC for holes (red circles), **e** Au for holes and Ca for electrons (blue circles) vs. Au/pc-BP2T/TTC for both holes and electrons (red circles). **f** Surface morphology of sc-RU, pc-RU/sc-RU, TTC/sc-RU, and pc-RU/TTC/sc-RU, respectively, measured with an atomic force microscope

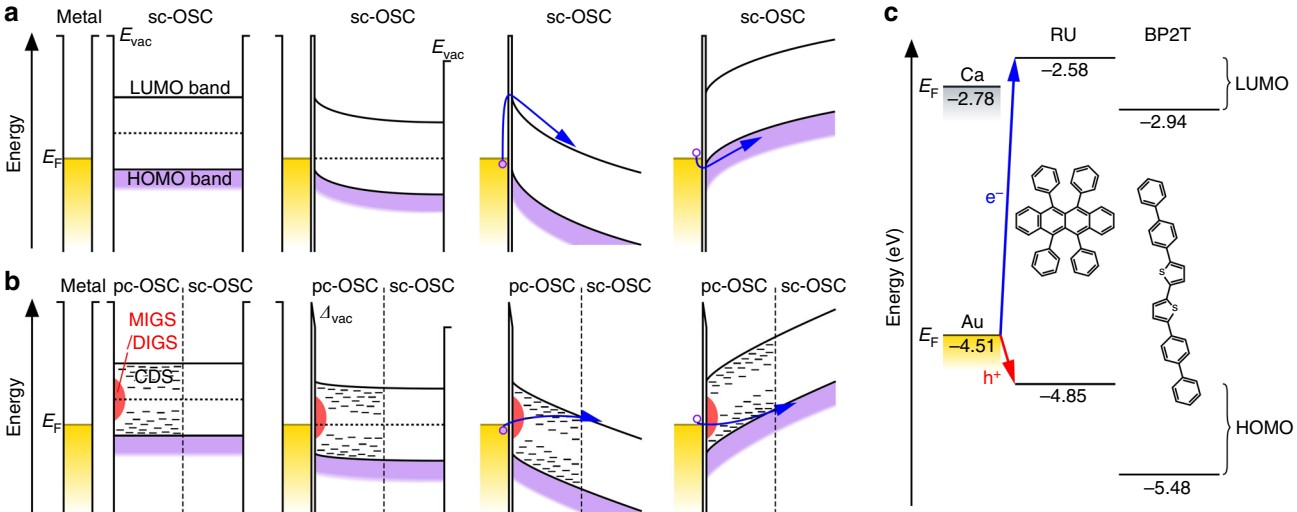

**Fig. 2** Energy diagrams. Carrier injection mechanism of **a** metal electrode of Schottky limit and **b** new electrode consisting of a bilayer (polycrystalline organic semiconductor (pc-OSC)/tetratetracontane (TTC) covered by a metal thin film (Au, Ca etc.) for FETs of single-crystalline OSC (sc-OSC). From left to right: separated metal and semiconductor, contacted metal and semiconductor, electron injection, and hole injection. **c** Fermi level ($E_F$) of Ca and Au, HOMO and LUMO of rubrene (RU) and 5,5′-di(4-biphenylyl)-2,2′-bithiophene (BP2T)

were fabricated by laminating a thin sc-OSC (rubrene (RU) or 5,5′-di(4-biphenylyl)-2,2′-bithiophene (BP2T), Fig. 2c) on a SiO$_2$/p$^{++}$-Si substrate. Because a protective layer is known to be necessary to prevent electron traps on a SiO$_2$ surface[21–24], the SiO$_2$ surface was coated with polystyrene. Electrodes were made by a successive deposition of TTC, pc-OSC, and Au or Ca layer on a sc-OSC.

We can consider two types of states inside the band gap of the pc-OSC, and each of them plays very important roles in a different manner in the new electrodes of M/pc-OSC/TTC and can offer higher ambipolar injection than any other electrodes so far reported. One is the gap states induced by the surface disorder of semiconductors (DIGS) or the MS interface (MIGS), and the other is the continuously distributed gap states (CDS) created by structural disorder of molecules in bulk pc-OSC. In the former, the high density of states inside the gap play an important role for pinning, giving the Bardeen limit[28]. As for the possible conversion from the Schottky to the Bardeen limit, we previously reported that a large vacuum level shift can occur at the MS interface, when the surface texture of pc-OSC is controlled by TTC, which is consistent for the Fermi level pinning associated with DIGS/MIGS[4]. In addition, CDS capture carriers regardless of their polarity and then release them into transporting channels as illustrated in Fig. 2b.

We optimized the thickness of a thermally evaporated pc-OSC on a sc-OSC active layer (Fig. 1b). In addition, we further studied a bilayer configuration of pc-OSC (20 nm)/TTC (5 nm) as the optimal interlayer, where the pre-deposited ultrathin TTC layer controls the disorder in pc-OSC. The change in morphology of pc-RU controlled by a TTC layer is demonstrated by atomic force micrographs in Fig. 1f. The average grain size of pc-RU was smaller for pc-RU on TTC/sc-RU than that for pc-RU on sc-RU.

**Carrier injection and field-effect mobility**. When a new electrode of M/pc-OSC/TTC (M=Au for holes and M=Ca for electrons) is compared to the hetero electrodes of Au (for hole injection) and Ca (for electron injection), much higher injection efficiencies for both polarity carriers were observed as shown in Fig. 1c. The advantage of the new electrode becomes clear when Au and Ca were employed for electrons and holes, respectively.

The new electrode still showed efficient carrier injection both for holes and electrons, while almost negligible carriers were injected from pure-metal electrodes (Fig. 1d). Even if air-stable new electrodes of Au/pc-OSC/TTC are used both for electrons and holes, the efficiency is importantly still comparable to the hetero electrodes of Au (for hole injection) and Ca (for electron injection) as shown in Fig. 1e. This is a very intriguing and seemingly counterintuitive result.

The two-terminal field-effect mobilities evaluated from transfer characteristics of FETs, which have two additional voltage probes for four-terminal measurements, were highly dependent on electrodes as plotted in Fig. 3a, b. Meanwhile, the differences of four-terminal mobilities among FETs were much smaller than those of two-terminal mobilities, and all two-terminal mobilities were lower than four-terminal mobilities as summarized in Supplementary Fig. 1. This indicates the differences in two-terminal mobilities are mainly due to the differences in contact resistances. With this in mind, three important features can clearly be deduced from Fig. 3a, b. First, the new electrodes of M/pc-OSC/TTC (magenta and cyan squares for M=Au and Ca, respectively, located in the top right corner) are equivalently effective for both hole and electron injection, being remarkably different from the conventional electrodes so far proposed. Second, increase in field-effect mobility by employing the new electrodes is significantly high, especially when the $E_F$ of metal electrode is far from CBM for electron injection or VBM for hole injection, i.e., Au for electron and Ca for hole. The mobilities increased compared to those of conventional metal electrodes by several orders of magnitude, which is indeed unexpectedly large. Third, the highest mobility for ambipolar injection was achieved with a combination of new electrodes of M/pc-OSC/TTC, where M=Au and Ca for hole and electron injection, respectively. The hole and electron mobilities of sc-RU FETs reached 22.0 and 5.0 cm$^2$ V$^{-1}$ s$^{-1}$, respectively. For sc-BP2T FETs, hole and electron mobilities were 0.5 and 1.1 cm$^2$ V$^{-1}$ s$^{-1}$, respectively. To our knowledge, these values are the highest among the two-terminal filed-effect mobilities of p- and n-type so far reported for sc-BP2T[6, 21] and sc-RU FETs[25, 26], which are plotted in combination as open diamonds in Fig. 3a, b. These have unambiguously been confirmed by many repeated experiments in the present work.

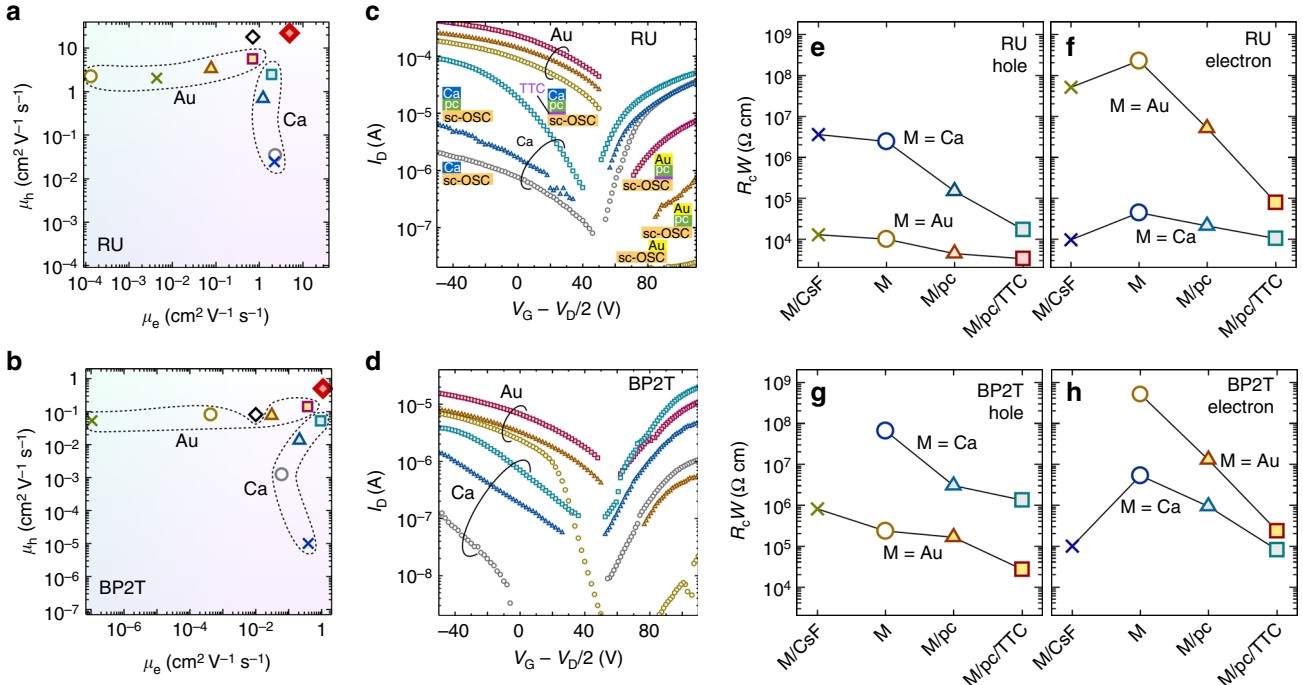

**Fig. 3** Transistor characteristics and contact resistances of single-crystalline organic semiconductors. Two-terminal field-effect mobilities of **a** rubrene (RU) and **b** BP2T single-crystal OFETs with various electrodes. Yellow-to-magenta marks indicate Au electrodes, and gray and blue-to-cyan marks indicate Ca electrodes: Open circles, closed triangles, closed squares, and crosses indicate no interlayer, pc-OSC interlayer, pc-OSC/TTC interlayer, and CsF interlayer, respectively. Red closed diamonds represent the highest mobilities obtained for FETs employing Au/pc-OSC/TTC and Ca/pc-OSC/TTC asymmetric electrodes (the present work). Open diamonds represent the highest mobilities of BP2T[6, 21] and rubrene[25, 26] single-crystal FETs reported in literature so far. Transfer characteristics of **c** RU and **d** BP2T OFETs respectively with various electrodes. Contact resistances of the source electrodes ($R_c$) normalized by the channel width ($W$) of various electrodes for **e**, **f** RU and **g**, **h** BP2T for hole and electron injection

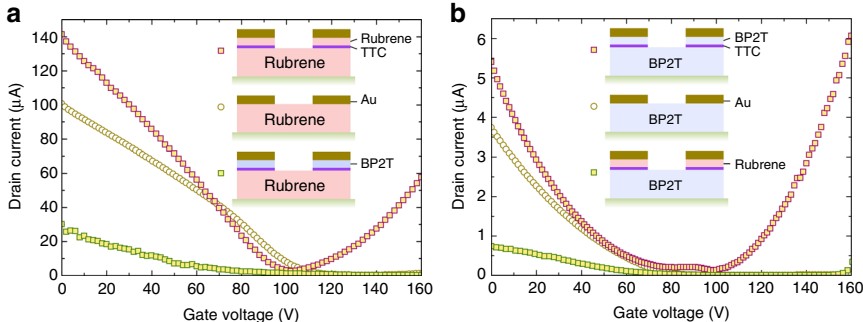

**Fig. 4** Transfer characteristics of organic single-crystal field-effect transistors with and without interlayer. **a** RU single crystal with Au electrodes (yellow circle), with Au/RU/TTC electrodes (magenta square), and Au/BP2T/TTC electrodes (green square). **b** BP2T single crystal with Au electrodes (yellow circle), with Au/BP2T/TTC electrodes (magenta square), and Au/RU/TTC electrodes (green square)

**Transfer characteristics and contact resistance**. It is intriguing to confirm again that electrons from Au/pc-BP2T/TTC (generally Ca is suitable) and holes from Ca/pc-BP2T/TTC (generally Au is suitable) can efficiently be injected as shown in the center panel of Fig. 1d. The data of the conventional electrodes plotted with blue circles, showing almost no carrier injection because of the unfavorable $E_F$ of the electrode for the carrier polarity as shown in Fig. 2a, c, but the situation was greatly improved when the new electrodes were used. The reason can be explained by a significant reduction in the barrier height due to CDS in pc-OSC as illustrated in Fig. 2b. A Fermi level shift toward the middle of the band gap at the MS interface due to the MIGS/DIGS is also important so that the equivalent barrier heights can be made between electron and hole injection. Reductions in injection barrier height for both holes and electrons was confirmed by

measuring temperature dependence of contact resistance using FETs with two voltage probes. Barrier heights of Au/pc-OSC/TTC for electron injection and Ca/pc-OSC/TTC for hole injection were comparable to those of pure Ca and Au electrode, respectively (Supplementary Fig. 2).

Detailed information can be obtained from transfer characteristics of RU and BP2T FETs with M, M/pc-OSC, and M/pc-OSC/TTC as electrodes, as shown in Fig. 3c, d. Evidently, the new electrodes can greatly enhance the drain current and decrease the threshold voltage, both of which are known to be largely associated with the reduction in contact resistance. Actually, contact resistances evaluated from four-terminal measurements were greatly reduced by inserting a pc-OSC and pc-OSC/TTC layer, as shown in Fig. 3e–h. For RU, the hole injection increases in the following order: Ca/CsF, Ca (without interlayer), Ca/pc-

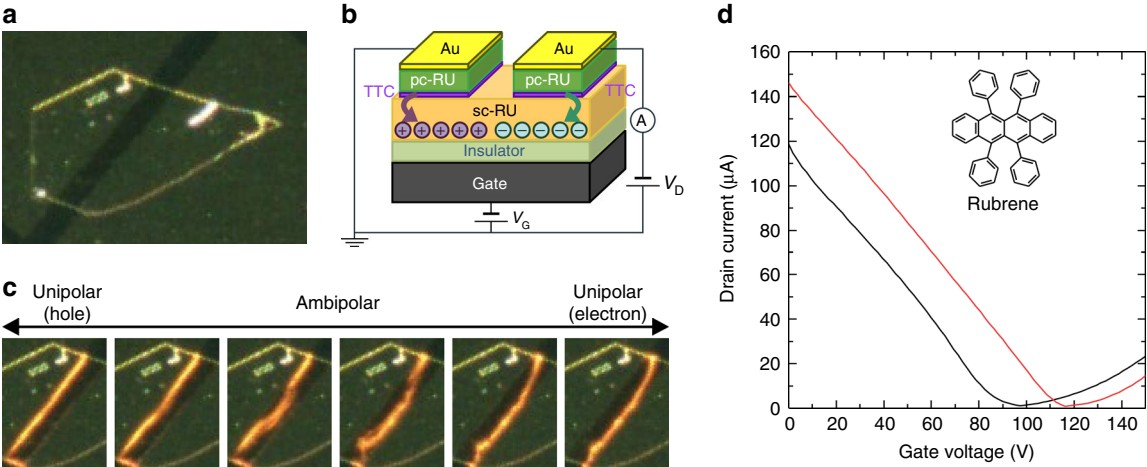

**Fig. 5** Light emission from a FET of a rubrene single crystal. **a** An optical micrograph. **b** A schematic illustration of a rubrene single crystal (sc-RU) FET. **c** Change in position of light-emitting zone with respect to the gate voltage ($V_G$), indicating ambipolar light emission. **d** Transfer characteristics of the sc-Ru FET before (black) and after (red) air exposure of 2 h. The drain voltage $V_D$ was 150 V

OSC, Ca/pc-OSC/TTC, Au/CsF, Au (without interlayer), Au/pc-OSC, and Au/pc-OSC/TTC. In the electron-injection/transport region, the injection was improved in the order of Au, Au/CsF, Au/pc-OSC, Au/pc-OSC/TTC, Ca≈Ca/pc-OSC, and Ca/CsF≈Ca/pc-OSC/TTC. In the case of BP2T, the hole injection was improved in the following order: Ca/CsF, Ca, Ca/pc-OSC, Ca/pc-OSC/TTC, Au/CsF, Au, Au/pc-OSC, and Au/pc-OSC/TTC, and the order for electron injection was Au/CsF, Au, Au/pc-OSC, Ca, Ca/pc-OSC, Au/pc-OSC/TTC, Ca/CsF, and Ca/pc-OSC/TTC. The same orders were obtained from output characteristics (Supplementary Figs. 3 and 4). The best performance was achieved with Au/pc-OSC/TTC and Ca/pc-OSC/TTC in hole and electron injection, respectively, for both RU and BP2T FETs. Note that the previously proposed CsF modification layer on the OSC surface is effective only for electron injection, and the efficiency deteriorates for holes as seen in Fig. 3. In contrast, the new electrodes of M/pc-OSC/TTC and M/pc-OSC work very efficiently regardless of the charge polarity thanks to the injection with low energy excitation via CDS as well as equivalently both for holes and electrons due to DIGS/MIGS.

To provide useful information on CDS generated inside the band gap of pc-OSC, we compared FETs employing two different pc-OSC materials, one of which is the same as sc-OSC and the other is different from it (Fig. 4). Therefore, the combinations are (1) Au/pc-RU/TTC and Au/pc-BP2T/TTC on sc-BP2T, and (2) Au/pc-RU/TTC and Au/pc-BP2T/TTC on sc-RU. Intriguingly, greatly better ambipolar injection was observed for Au/pc-RU/TTC on sc-RU and Au/pc-BP2T/TTC on sc-PB2T than that of Au on sc-RU and sc-BP2T, while much worse carrier injection was observed for Au/pc-BP2T/TTC on sc-RU and Au/pc-RU/TTC on sc-BP2T. These results unambiguously demonstrate that CDS is created inside the band gap and the matching of CDS in pc-OSC to CBM and VBM of sc-OSC is crucial. Carriers can be injected from the $E_F$ of a metal thin film via CDS with low-excitation energy and are smoothly transferred to an active transporting sc-OSC layer through a porous TTC layer via tunneling through TTC or direct injection to a sc-OSC.

**Light-emitting FET with air stable electrodes.** A LE-FET was made by employing the new concept of air stable electrode Au/pc-OSC/TTC for RU as shown in Fig. 5. A bright emission with the current density of 25 kA cm$^{-2}$ surpassing the highest reported value[16] of 4 kA cm$^{-2}$ for organic single-crystalline LE-FETs without processing the crystal was obtained for RU. Because

the new electrodes are air stable, the LE-FET device worked even after exposure to air as shown in Fig. 5d.

**Discussion**

A new concept of electrode of M/pc-OSC/TTC for carrier injection into sc-OSCs offers not only higher hole injection than that of a pure Au electrode but also higher electron injection than that of a pure Ca electrode. In addition, almost equivalent efficiencies of hole and electron injection are obtained with M/pc-OSC/TTC (M=Au or Ca).

There are several conceivable mechanisms of the low contact resistances in the new electrode for both hole and electrons. The mechanisms are divided into two types. One relats with morphology change of the MS interface, and another relates electronic property change of it. The former possibility disagree with experimental results that a pc-RU and pc-BP2T interlayer decrease contact resistance only for sc-RU and sc-BP2T, respectively, and they are not interchangeable as shown in Fig. 4. Therefore, the lowering of contact resistances is attributed to the latter type, which is further divided into three cases considering that the contact resistances are caused by Schottky barriers shown in Fig. 2a.

First, the height of a Schottky barrier is varied by shift of $E_F$ of the metal electrode with respect to the CBM and VBM. Second, the width of a Schottky barrier is thinned by carrier doping to pc-OSC. Third, carriers are injected by means of hopping transport through states in a band gap. In the first case, although a shift of $E_F$ decreases a Schottky barrier height for holes or electrons, it increases a Schottky barrier height for the other carrier at the same time. Therefore, although it can help to balance hole and electron injection, it cannot explain decrease of contact resistances for both electron and holes. In the second case, although electron and hole doping can reduce a Schottky barrier width for electron and hole injection, respectively, the doping increases a contact resistance for the other carrier. Therefore, this case is also inconsistent with decreasing of both hole and electron contact resistance. Therefore, the third case can only explain the present experimental results.

As electronic states in a band gap, highest occupied molecular orbitals and lowest unoccupied molecular orbitals of pc-OSC, which deviate from a valence and conduction band, respectively, by structural disorder, are most likely origin. Such gap states are widely observed in pc-OSC with temperature dependent transport measurements[29] and ultraviolet photoelectron

spectroscopy[30], and enhancement of carrier injection by increasing gap states was reported for a pc-OSC thin film with a combined study of ultraviolet photoelectron spectroscopy and transport measurements[31].

By employing the electrodes proposed based on the new concept, the highest two-terminal electron and hole mobilities as far as we know is realized. It is noted that the new electrode can be applicable to various OSCs and is not limited only to RU and BP2T, because the mechanism of carrier injection is very general and is not greatly affected by the properties of OSCs. The new electrode structure perfectly works in the top-contact FET structure, and it is also, in principle, applicable when electrostatic laminating of a sc-OSC on a pc-OSC layer cannot cause serious problems even in the case of bottom-contact and top-gate FET structure. The new concept of electrodes for OSCs is very promising for air-stable high performance EL devices and electronic logic circuits as well as organic lasers in the future.

## Methods

**Materials for interlayers**. The purity of the interlayer compound (pc-OSC) is very important for the electron injection in the present studies. Therefore, a thin-film FET of pc-OSC with Au electrodes was fabricated to confirm an equivalent ambipolar performance, which indicates a high purity of pc-OSC material. First, a TTC (9 nm) thin film was deposited on $SiO_2/p^{++}$-Si substrate, which was cleaned by ultrasonication in acetone, ethanol, and 2-propanol followed by $O_2$-prasma treatment, followed by thin-film growth of pc-OSC (30 nm) and finally deposited Au source and drain electrodes with a thermal vacuum evaporator[4]. Purification of pc-OSC and fabrication of an FET were repeated until the FET gives an equivalent ambipolar injection. After the confirmation, the purified pc-OSC was used as a source material for interlayers. During this process, the pc-OSC material was never exposed to air until the electrode deposition.

**Device fabrication**. Thin films of sc-OSCs were grown using a physical vapor transport method using RU (≥98%) and BP2T (97%) purchased from Sigma-Aldrich Co. under Ar gas flow in a quartz tube heated in a tube furnace with a temperature gradient. The crystal growth was repeated for three times to purify the materials. A $p^{++}$-Si substrate with a $SiO_2$ layer of 300 nm in thickness was spin-coated with a 1 wt% solution of polystyrene in toluene. The thickness of the polystyrene film was 25 nm. The substrate was dried in Ar atmosphere at 75 °C for more than one night, and a sc-OSC was laminated on it. Materials of interlayers, i.e., TTC, pc-RU, pc-BP2T, and CsF, and Au or Ca electrodes were deposited employing a stencil mask with a vacuum thermal evaporation method. The thicknesses of the TTC and pc-OSC layers were 5 and 20 nm, respectively. Performances of FETs are very sensitive to the thickness of TTC layer. If the TTC thickness deviated from the optimal thickness, the device performance drastically decreased.

**Measurements**. All electrical measurements were carried out with a B1500A Semiconductor Device Parameter Analyzer (Keysight Technologies Inc.) in a glove box under Ar gas atmosphere at room temperature. Surface morphologies of sc-RU, pc-RU/sc-RU, TTC/sc-RU, and pc-RU/TTC/sc-RU were measured with an atomic force microscope (Keysight Technologies Inc.) in air.

**Data availability**. The data that support the findings of this study are available from the corresponding authors upon request.

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

## Acknowledgements

This work was partially supported by JSPS KAKENHI Grant Number JP17H05326. H.S. acknowledges JSPS KAKENHI Grant Number JP24684023, JP25610084, and JP16K13826, JGC-S Scholarship Foundation, and CASIO Science Promotion Foundation. T.K. and K.T. are grateful for financial support for collaborative researches by AIMR, Tohoku University.

## Author contributions

The research idea and design of experiments for its confirmation were made by T.K., H.S. and K.T. FET experiments were performed by T.K. and AFM measurements were carried

out by R.S. and T.H. The experimental data were analyzed by T.K., H.S. and K.T., T.K., H.S., and K.T. prepared the manuscript.

## Additional information

**Competing interests:** The authors declare no competing financial interests.

