## [Peer Review File · Nature Communications]

Reviewers' comments:

Reviewer #1 (Remarks to the Author):

I have problems with this manuscript that prevent me from recommending its publication anywhere let alone in Nature Comm.

The authors seem to have a misconception of what the Bardeen limit is, how it relates to Schottky barriers, and Fermi level pinning. This misconception should not be further propagated in the literature. They seem to think that in gap states are a good thing, that somehow explain the performance of their devices, but their picture is seriously flawed. Fermi level pinning refers to the observation that many metal/inorganic semiconductor (ISC) interfaces do not follow the Schottky-Mott model in predicting the correct barrier characteristics as being related to the metal work function. Bardeen's ansatz was that in those cases the charge exchange and Fermi level equilibration responsible for the Schottky barrier formation, rather than occurring with the bulk semiconductor states, instead occurs with a pocket of the semiconductor surface states that possess a high density of electronic states. Thus the Fermi level of the ISC is seemingly pinned and does not respond to the metal work function (the barrier characteristics are rather dictated by an equilibration that has already occurred between the bulk and its surface states). That pocket of high electron density surface states frequently lies somewhere in the gap so these are often called gap states but that does not mean that they form some convenient channel to inject charge into the semiconductor as indicated in their Figure 1b.

Many researchers have used interfacial layers to populate those surface states or to create dipoles at the interface that effectively shift the metal work function, both improving injection. For disordered interfaces alternative models have been proposed like the potential fluctuation model for rough surfaces (J. H. Werner and H. H. Güttler, *J. Appl. Phys.* 69, 1522, 1991) or size effects on mixed phase contacts where the disorder is on the scale of the depletion width (J. L. Freeouf, T. N. Jackson, S. E. Laux, J. M. Woodall, *Appl. Phys. Lett.* 40, 634, 1982) but I am not aware of any accepted theoretical treatment that fits their picture.

A more likely explanation for the phenomena they observe is that, as shown in their Figure 2c, their polycrystalline organic semiconducting (pc-OSC) layer is quite rough. This leads to a texturing of the metal electrodes that leads to sharp protrusions of the Au down to, or almost to, the TTC layer. There is generally a field concentration that occurs at sharp metal points that increases the local field strength. So what they are likely observing is a field emission phenomenon, in which the high local field strength thins the barrier sufficiently to inject electrons from the Au. Note the large positive gate voltage, against the negative drain required to achieve the injection. To demonstrate that my alternative picture is incorrect they should be able to use an amorphous OSC that forms a far less rough electrode, but which should contain even more of the "gap states" (within what I claim is their flawed picture) to still achieve their ambipolar injection from an Au electrode. I am generally loath to ask for more experiments in a review but in this case I would have to see such an experiment to be convinced that they have anything beyond what I've described.

That said I am also sensitive to the idea that there are controversial findings that sometimes herald new ideas and that the debate about these are better held in the literature, in public view, rather than by a squelching of the result at the review stage. In that vein if the authors can, in a rewrite, provide much better support for what I call their flawed interpretation of the Bardeen limit, or provide an alternative that better fits accepted models and/or address my alternative scenario to their data to pass another round of reviews then perhaps their manuscript can become publishable.

Reviewer #2 (Remarks to the Author):

This work presents a new concept for highly efficient carrier injection into organic semiconductors using two buffer layers of TTC and pc-OSC. Although the insertion of buffer layer is not new idea, the carrier injection efficiency of this new type electrode is almost equivalent both for hole and electron, which is impossible for previous types of electrodes. The results are very clear and the effectiveness of new electrode is obvious.

The main weakness of the present work lies in that the detail measurement to prove the mechanism is seriously lacking. After answering this comments, I will agree with publication of this manuscript in Nature Commun.

1. Author should clarify the induced IGS by pc-OSC layer using experimental way, because this is most important point of this research. Although authors explains the possible mechanism, most part is not supported by experimental results.
2. Author should show more experimental results, such as out-put characteristics, the mobility difference between two-terminal and four-terminal measurements, and transistor characteristics before and after air exposure. Particularly, if author want to claim air stability, comparison before and after air exposure is key results.
3. The evaluation of injection efficiency is not quantitative. The temperature dependence of IV characteristic is one possible method to evaluate injection efficiency.
4. I think the arguments on two terminal mobility (comparisons and the statement of high mobility) is not important because the main point of this research is injection efficiency and two terminal mobility is also dependent on crystal quality. Therefore, contact resistance is key parameter. However, the contact resistance shown in Fig. 3c is not small, comparing with the previous champion report. Please explain the reasons.
5. The effect of induced damage during deposition process is not negligible. The comparison between top and bottom contact is another important approach.
6. The statement of "bright emission" without quantitative results is not fair. Therefore, if author want to claim "bright emission", the EQE of fabricated device is necessary. Moreover, I am not sure that it is really ambipolar light emission because channel length is too short. The demonstration with longer channel transistor is also required.

Reviewers' comments:

Reviewer #1 (Remarks to the Author):

While I remain skeptical about the mechanism proposed by the authors, I believe the phenomena they report is interesting, potentially important (given the multiple systems in which it operates) and merits further investigation by the broader community. I do wish that they addressed, in the revised manuscript, the possible role that field emission and tunneling play in what is observed (including arguments that in their view rule this out). Nevertheless, their revision does a better job of laying out their proposed mechanism so that it can be more readily understood, and if they are correct, it will be an important result. I therefore lift my previous objection to publication in Nature Communications. I do note one item that they must fix in their supporting information Figure S3. They label the horizontal axis of their output curves as "Gate Voltage". This should be Drain Voltage.

Reviewer #3 (Remarks to the Author):

After carefully reading the manuscript as well as the response to the comments of previous two reviewers, I believe that the ms is potentially interesting to the target audience of Nature Communications. Especially, the authors have followed the constructive comments of Reviewer 2 to further explain the mechanisms.

However, I still have one concern: How general is this new electrode approach is? Does it only work for some specific cases (e.g. BP2T6 and rubrene as shown by the authors)? Or is it a general approach which works for OFETs? For the later case, I would suggest to accept the ms. Otherwise, I would think that it fits better to a specialized journal.

Reviewer #4 (Remarks to the Author):

The paper „New concept of electrode in organic semiconductors“ by Thangavel Kanagasekaran et al reports a novel approach to contact organic single crystals with a polycrystalline layer, SAM, and metal electrode. The data clearly show that these contacts work very well and are a significant advance compared to previous approaches, although one of the claims, that ignoble metals can be used, is not really fulfilled since Ca is still present. Despite the excellent experimental work on many different transistor structures, the underlying physics is limited. In particular, an experimental proof for the main claim und key underlying mechanism, the continuous disordered states (CDS), e.g. by spectroscopy, is not given. I thus believe that the paper does not really has the level needed for Nature Communications. Maybe „Scientific Reports“ would fit. Furthermore, I would add two issues before the paper is published:

- A very thorough language polish is needed: As example, I cite the following sentence „This has been one of major problems for a long year that injection of electrons is greatly difficult compared to those of holes in OSCs since stable metals with low E_F , such as Au, Cu, Ag, are typically employed.“

- I find the sentence p.4 l.50 „Such ambipolar injection and subsequent light emission have attracted much attention from the viewpoint of optoelectronics since no chemically doped pn junction is required for light-emitting field-effect transistors (LE-FET)“ confusing, although todays commercial OLED are often chemically doped, there have been many which are not.

Replies to the comments by the 1st reviewer

First, we thank the reviewer for careful reading of our manuscript and the comments on the Bardeen limit. We think that our insufficient explanation in Page 2 and Fig.1 gave a wrong impression to the reviewer. Importantly, we consider two types of states inside a band gap, and each of them makes very important roles in a different manner for the new electrodes of M/pc-OSC/TTC and can offer the higher ambipolar carrier injection than any other electrodes so far reported. One is the gap states induced by the surface disorder of semiconductors (disorder induced gap states) or the metal-semiconductor interface (metal-semiconductor interface induced gap states), and the other is the gap states created by disorder of molecules in the bulk polycrystal semiconductors continuously spreading inside the band gap of semiconductors (continuous disordered states, CDS). In the former case, the high density of states inside the band gap plays at the interface a very important role for pinning, giving the Bardeen limit in terms of the vacuum level shift at a semiconductor surface (J. Bardeen, Surface States and Rectification at a Metal Semi-Conductor Contact, Physical Review **71** 717 (1947)). The latter gives the states to allow for occupation of both types of carriers, holes and electrons regardless of the charge polarity, and for release to other states without trapping for causing good electrical transport and efficient ambipolar carrier injection. The alternative interpretation of “*a field emission phenomenon, in which the high local field strength thins the barrier sufficiently to inject electrons from the Au*” based on the Bardeen Ansatz frequently used in the mechanism of STM commented by the reviewer cannot be applicable in the new concept of electrodes. We have made useful experiments to be able to settle down this argument and explained the details in the following paragraphs.

To answer the question of an alternative possible mechanism of “*a field emission phenomenon, in which the high local field strength thins the barrier sufficiently to inject electrons from the Au*” commented by the reviewer, first we explain using our experimental data reported previously in Applied Physics Letters (T. Kanagasekaran *et al.*, **107**, 043304 (2015)) and describe our additional experiments newly made for this purpose, which have been added as Fig. S5 of Supplemental Information in the revised manuscript for general readers. We think that the reviewer will like the new experimental results.

As explained in APL (T. Kanagasekaran *et al.*, *Appl. Phys. Lett.* **107**, 043304 (2015)), the morphology of polycrystal BP2T (pc-BP2T) was changed by employing poly(methylmethacrylate) and tetratetracontane (TTC) for a substrate modification layer. The former was used for growing a better regulated pc-BP2T by vapor deposition, and the latter is used for making disordered pc-BP2T. As we reported earlier, the better ambipolar carrier injection can be observed for the disordered pc-BP2T as shown in Fig. I, and this became an important hint to have an idea of the new electrode described in the present study. At the same time, we clarified band diagrams at metal-semiconductor (MS) interfaces using photoelectron spectroscopy. Important experimental evidences were achieved as shown in Fig. II. In the case of regulated pc-BP2T, the vacuum level shift at the MS interface was nearly zero. On the other hand, pc-BP2T having more disordered texture grown on a TTC layer showed a large vacuum level shift, which is consistent with the change from the Schottky to the Bardeen limit in MS contact scheme.

Figure I | The output characteristics of a BP2T FET with Au-Au homoelectrodes. (a) on TTC-SiO₂ and (b) on PMMA-SiO₂.

To convince the reviewer further, we have made additional new experiments (Fig. S5). By keeping the same optimum condition of experiments, we compared a different pc-OSC from the active OSC as a new electrode: (1) Au/pc-RU/TTC and Au/pc-BP2T/TTC on sc-BP2T and (2) Au/pc-RU/TTC and Au/pc-BP2T/TTC on sc-RU. Intriguingly, much better ambipolar carrier injection can be observed for Au/pc-RU/TTC on sc-RU and Au/pc-BP2T/TTC on sc-PB2T than that of Au on sc-RU and sc-BP2T, while much worse carrier injection was observed for Au/pc-BP2T/TTC on sc-RU and Au/pc-RU/TTC on sc-BP2T. These experimental data unambiguously give the important fact that CDS created in bulk pc-OSC play an important role for reducing the barrier height for carrier injection followed by tunneling through the ultrathin-TTC layer to the active transporting layer of sc-OSC. Definitely, the energy level matching between pc-OSC and sc-OSC is crucial for better carrier injection, indicating that the extremely good ambipolar carrier injection is mainly caused by CDS.

Figure II Band diagram of Au and BP2T in eV without contact (left), and a Au-BP2T interface on PMMA-SiO₂ (center) and TTC-SiO₂ (right). E_F and Δ denote the Fermi level and the vacuum level shift at the interface, respectively. The lines inside the red zone in color indicate the gap states.

Figure S5 | Transfer characteristics of organic single-crystal field-effect transistors with and without interlayer. (a) Rubrene single crystal with Au electrodes (yellow circle), with Au/rubrene/TTC electrodes (magenta square), and Au/BP2T/TTC electrodes (magenta square). (b) BP2T single crystal with Au electrodes (yellow circle), with Au/BP2T/TTC electrodes (magenta square), and Au/rubrene/TTC electrodes (magenta square). (c) BP3T single crystal with Au electrodes (yellow circle), with Au/BP2T/TTC electrodes (violet square).

These two experiments clearly show that the change in the carrier injection limit from the Schottky to the Bardeen due to the high-density pinning states and the carrier injection with low energy excitations via CDS created in pc-OSC are the most important factors for efficient ambipolar carrier injection. Although the Bardeen-Ansatz in tunneling, generally being importantly considered for the mechanism of STM, may play some roles, but this gives a minor influence on ambipolar carrier injection in the new electrode for OSC field-effect transistors.

We have provided here some important experimental information for replying to the comments raised by the reviewer. The two types of levels are created inside a band gap, one of which is DIGS/MIGS acting as the trapping level accompanied by the vacuum level shift in the Bardeen limit and the other is CDS providing the states to greatly enhance the carrier injection with a low energy barrier height. In order not to mislead the readers in their understanding we have modified the Fig.1 and have revised the explanations in Page 3 (shown in yellow). We hope that the reviewer understands the mechanism and will like our additional experiments. We again thank the reviewer for his spending a valuable time for the reviewing process.

Replies to the comments by the 2nd reviewer

Reviewer #2 (Remarks to the Author):

This work presents a new concept for highly efficient carrier injection into organic semiconductors using two buffer layers of TTC and pc-OSC. Although the insertion of buffer layer is not new idea, the carrier injection efficiency of this new type electrode is almost equivalent both for hole and electron, which is impossible for previous types of electrodes. The results are very clear and the effectiveness of new electrode is obvious. The main weakness of the present work lies in that the detail measurement to prove the mechanism is seriously lacking. After answering this comments, I will agree with publication of this manuscript in Nature Commun.

Thank you very much for your taking valuable time for the reviewing process of our manuscript. We are very pleased to hear your comments for agreeing with publication in Nature Communications, after replying your comments. We have replied to the comments and revised our manuscript by referring to the comments as follows:

1. Author should clarify the induced IGS by pc-OSC layer using experimental way, because this is most important point of this research. Although authors explains the possible mechanism, most part is not supported by experimental results.

[Reply]

The difference in carrier injection between the Schottky and Bardeen limit can experimentally be recognized by the fact that the Schottky barrier is strongly dependent on the work function of metal electrodes (ϕ_M) for the Schottky limit, while it is nearly independent of ϕ_M for the Bardeen limit. As shown in Fig. III taken from our previous report (T. Kanagasekaran et al., *Appl. Phys. Lett.* **107**, 043304 (2015)). When less disordered polycrystal BP2T (pc-BP2T) was grown on a PMMA/SiO₂ substrate, the Schottky limit was observed, while the more disorders created by employing a TTC/SiO₂ substrate can modify it to the latter Bardeen limit. The vacuum level shift at the metal-semiconductor interface was negligible in the former case and consequently the barrier height against carrier injection is determined by the difference between ϕ_M and conduction band minimum and valence band maximum for electron and hole injection, respectively. On the other hand, there was a considerable vacuum level shift in the latter case indicating a significant deviation from the Schottky limit. The experimental results support the fact that the gap states at the metal-semiconductor interfaces are different between the two cases.

The experimental results support the fact that the gap states at the metal-semiconductor interfaces are different between the two cases.

To provide additional experimental evidence for the continuous disordered states (CDS) generated inside the band gap of pc-OSC, we have made additional experiments (Fig. S5). By holding the same optimum condition of experiments, we compared a different pc-OSC from the active sc-OCS as a new electrode: Au/pc-RU/TTC and Au/pc-BP2T/TTC on sc-BP2T and Au/pc-RU/TTC and Au/pc-BP2T/TTC on sc-RU. Intriguingly, good ambipolar carrier injection can be observed for Au/pc-RU/TTC on sc-RU and Au/pc-BP2T/TTC on sc-BP2T than that of Au on sc-RU and sc-BP2T, while much worse carrier injection was observed for Au/pc-BP2T/TTC on sc-RU and Au/pc-RU/TTC on sc-BP2T. These experimental data unambiguously give the fact that the CDS created in bulk pc-OSC play the important role for reducing the barrier height for carrier injection to sc-OSC via tunneling through the ultrathin TTC layer. The energy level matching between pc-OSC and sc-OSC is crucial.

Figure III | Band diagram of Au and BP2T in eV without contact (left), and a Au-BP2T interface on PMMA-SiO₂ (center) and TTC-SiO₂ (right). E_F and Δ denote the Fermi level and the vacuum level shift at the interface, respectively. The lines inside the red zone in color indicate the gap states.

Figure S5 | Transfer characteristics of organic single-crystal field-effect transistors with and without interlayer. (a) Rubrene single crystal with Au electrodes (yellow circle), with Au/rubrene/TTC electrodes (magenta square), and Au/BP2T/TTC electrodes (magenta square). (b) BP2T single crystal with Au electrodes (yellow circle), with Au/BP2T/TTC electrodes (magenta square), and Au/rubrene/TTC electrodes (magenta square). (c) BP3T single crystal with Au electrodes (yellow circle), with Au/BP2T/TTC electrodes (violet square).

These two experiments clearly show that the change in the carrier injection limit from the Schottky to the Bardeen due to the high-density pinning levels and the carrier injection with low energy excitation via CDS created in the bulk pc-OSC are the most important factors for providing the extremely high ambipolar carrier injection. These are unambiguous and sufficient experimental evidence for supporting the mechanism of highly efficient ambipolar carrier injection.

2. Author should show more experimental results, such as out-put characteristics, the mobility difference between two-terminal and four-terminal measurements, and transistor characteristics before and after air exposure. Particularly, if author want to claim air stability, comparison before and after air exposure is key results.

[Reply]

According to the comments of the reviewer, we have prepared output characteristics, the mobility difference between two-terminal and four-terminal measurements, and the transfer characteristics before and after air exposure as Supplemental Information. Just in case for the reviewer, we also list the same data here.

Figure S1 | Four-terminal (cyan circles) and two-terminal (red squares) mobilities of single-crystalline organic semiconductor (sc-OSC) field-effect transistors with M/CsF, M, M/polycrystalline-OSC (pc-OSC) and M/pc-OSC/ tetratetracontane (TTC) electrode (M = Au or Ca). (a, b) P-type and n-type mobilities of rubrene. (c, d) P-type and n-type mobilities of 5,5'-di(4-biphenyl)-2,2'-bithiophene (BP2T). Reliable values were not obtained for some transistors because of large contact

Figure S3 | Output characteristics of single-crystalline Rubrene (sc-RU) field-effect transistors with M, M/pc-RU and M/pc-RU/ tetratetracontane (TTC) electrode (M = Au or Ca). (a) P-type characteristics. (b) N-type characteristics. Left panels: M electrode. Center panels: M/pc-RU electrode. Right panels: M/pc-RU/TTC electrode. Top panels: M = Au. Bottom panels: M = Ca.

Figure S4 | Output characteristics of single-crystalline 5,5'-di(4-biphenyl)-2,2'-bithiophene (sc-BP2T) field-effect transistors with M, M/pc-BP2T and M/pc-BP2T/ tetratetracontane (TTC) electrode (M = Au or Ca). (a) P-type characteristics. (b) N-type characteristics. Left panels: M electrode. Center panels: M/pc-BP2T electrode. Right panels: M/pc-BP2T/TTC electrode. Top panels: M = Au. Bottom panels: M = Ca.

Figure S6 | Transfer characteristics of organic single-crystal field-effect transistors before (black) and after (red) air exposure of 2 h. (a) Rubrene single crystal (sc-RU) with Au/polycrystalline rubrene/TTC electrodes. **(b)** BP2T single crystal (sc-BP2T) with Au/ polycrystalline BP2T/TTC

3. The evaluation of injection efficiency is not quantitative. The temperature dependence of IV characteristic is one possible method to evaluate injection efficiency.

[Reply]

To evaluate injection efficiency, we have carried out measurements of temperature dependences of contact resistances. From the experimental results, we obtained carrier injection barriers and summarized them in Fig. S2.

It is clearly shown that the pc-OSC layer decreases the injection barrier of both hole and electron, and the insertion of the TTC layer further decreases the injection barrier. The carrier injection barriers of M/pc-OSC/TTC of unfavorable metal M are comparable to those of favorable metal without pc-OSC/TTC layer.

Figure S2 | Activation energies of carrier injection to single-crystalline organic semiconductor (sc-OSC) field-effect transistors with M, M/pc-OSC and M/pc-OSC/tetratetracontane (TTC) electrode (M = Au or Ca). (a, b) Hole and electron injection into a rubrene single crystal. **(c, d)** Hole and electron injection into a rubrene single crystal. Activation energies of Ca electrodes for hole injection and a Au electrode for electron injection into BP2T could not be obtained because of too large contact resistances

4. I think the arguments on two terminal mobility (comparisons and the statement of high mobility) is not important because the main point of this research is injection efficiency and two terminal mobility is also dependent on crystal quality. Therefore, contact resistance is key parameter. However, the contact resistance shown in Fig. 3c is not small, comparing with the previous champion report. Please explain the reasons.

[Reply]

The reported contact resistances, which are lower than those in the present report, are only obtained in the case of bottom-gate bottom-contact configuration, while the configuration in the present work is bottom-gate top-contact.

The contact resistances in the present report are the lowest among FETs with bottom-gate top-contact configuration. The difference of the two configurations is attributable to the distance between electrodes and accumulation layers. In the present work, we focused on the improvement in carrier injection at the metal-semiconductor interfaces without changing electrode configurations.

5. The effect of induced damage during deposition process is not negligible. The comparison between top and bottom contact is another important approach.

[Reply]

It is possible that the damage during metal deposition process give some effect. A small injection of holes from a pure Ca electrode and electrons from a pure Au electrode despite large Schottky barriers might have occurred through in-gap states induced by the damage. However, these injections are negligibly small compared with injections from the new electrodes. Therefore, the conclusion that a pc-OSC layer reduces a contact resistance did not change by the effect.

As the reviewer considered, we also thought that the comparison between top and bottom contact will give further detailed information, and have tried fabricating bottom-contact FETs. However, sc-OSCs were not stuck on the electrodes covered by pc-OSC as shown in the left panels of Fig. IV, and were damaged during the device fabrication process as shown in the right panels of Fig. IV. The application of the new electrodes for the device with the bottom-contact structure still needs further technical improvements.

Figure IV | Optical micrographs of sc-OSC on Au electrodes covered with pc-OSC.

6. The statement of “bright emission” without quantitative results is not fair. Therefore, if author want to claim “bright emission”, the EQE of fabricated device is necessary. Moreover, I am not sure that it is really ambipolar light emission because channel length is too short. The demonstration with longer channel transistor is also required.

[Reply]

The statement of “bright emission” is used just in comparison with single-crystal FETs of the same material with Ca and Au hetero-electrodes. The statement is rationalized by the fact of the large ambipolar current, only supposing that the EQE is comparable if the sc-OSCs used as a transporting and emitting layer are high quality.

As for the 2nd comment, to show a fact that the emission of ambipolar carriers occurs in the carrier transporting layer, we replaced the original figure by a new optical micrograph of light emission from a FET in Fig. 4 by those obtained from a FET with a longer channel. In this FET, emissions from a point away from the electrodes are clearly seen, which means the emission is an ambipolar light emission.

Fig.4| Light emission from a FET of a rubrene single crystal (sc-RU) with Au/pc-RU/TTC electrodes. Top panels: an optical micrograph (top view) and schematic illustration of a sc-RU FET. Lower panels: Change in position of light-emission zone and luminescence intensity with respect to the gate voltage (V_G).

Replies to the reviewer's comments:

Replies to the comments

%%%*****

Reviewer #1:

%%%---

1) While I remain skeptical about the mechanism proposed by the authors, I believe the phenomena they report is interesting, potentially important (given the multiple systems in which it operates) and merits further investigation by the broader community. I do wish that they addressed, in the revised manuscript, the possible role that field emission and tunneling play in what is observed (including arguments that in their view rule this out). Nevertheless, their revision does a better job of laying out their proposed mechanism so that it can be more readily understood, and if they are correct, it will be an important result. I therefore lift my previous objection to publication in Nature Communications.

%%%---

Thank you very much for your reading of our revised manuscript and your support of our manuscript for publication in Nature Communications. I believe that our new electrode for ambipolar carrier injection gives a broad interest to scientific communities and fulfills a deeper understanding of the mechanism of carrier injection among research communities.

%%%=== Comment2

2) I do note one item that they must fix in their supporting information Figure S3. They label the horizontal axis of their output curves as "Gate Voltage". This should be Drain Voltage.

%%%---

Thank you for your notice. We have made the corrections of the horizontal axes of Fig.S3 and Fig.S4.

%%%*****

Reviewer #3:

Comment#1 : ===

1) I believe that the ms is potentially interesting to the target audience of Nature Communications. Especially, the authors have followed the constructive comments of Reviewer 2 to further explain the mechanisms.

%%%---

Thank you for your reading of our manuscript and support for publication of our manuscript in Nature Communications.

Comemnt2 : ===

2) However, I still have one concern: How general is this new electrode approach is? Does it only work for some specific cases (e.g. BP2T6 and rubrene as shown by the authors)? Or is it a general approach which works for OFETs? For the later case, I would suggest to accept the ms. Otherwise, I would think that it fits better to a specialized journal.

%%%---

[Fig.R1 was redacted]

The new concept of the electrodes can be applied as a high performance electrode for a variety of organic semiconductors and not limited only to rubrene (RU) and BP2T. This is because the conversion of the Schottky to the Bardeen limit and the continuous disorder states (CDS) inside the gap can give high efficient carrier injection irrespectively of charge polarity and this influence is general and does not depend on materials. For the reviewer, we show another example of tetracene, one of the typical OSCs, in order to show how the new electrode is generally effective to other materials as shown in FigR1. Since the new electrode can be applicable to a variety of OSCs without being limited to RB and BP2T, the work deserves the publication in Nature Communications according to the reviewer's comments. For general readers, we have added this information using a few sentences colored in yellow.

%%%*****

Reviewer #4.

First, thank you very much for taking your important time for reading of our manuscript. However, we believe that many of the comments arise from misunderstanding of the reviewer. We reply and explain each by each so that the reviewer can have his/her correct understanding on our work.

%%%===

1). The data clearly show that these contacts work very well and are a significant advance compared to previous approaches, although one of the claims, that ignoble metals can be used, is not really fulfilled since Ca is still present.

%%%---

Reviewer wrote this comment using a complex sentence in a wrong fashion. We think that the reviewer would like to comment that "Ca is not noble metal but this is used in the new electrode. Then, why does the author claim that the new electrode is air stable." If this is the case, as described in the manuscript, the structure of the new electrode is M(metal thin layer)/p-OSC/TTC. Importantly the performance of the new electrodes is better than those of any other metals, such as Ca for electron and Au for hole injection, even when Au(noble thin film)/p-OSC/TTC is used. Au/p-OSC/TTC (not having ignoble electrode) is comparable or better than those of both Ca and Au. This is the indeed unusual and important result. Au/p-OSC/TTC is air stable and shows the highest performance among electrodes, which do not use any ignoble metals. These contents were included in the original manuscript and the stability in air was shown also in Supplemental Information (Fig.S6). We think that the reviewer did not recognize when

he/she read the manuscript.

Please keep in mind that “ignoble metal” is not a correct scientific word, although noble metal can be used as the definition: The **noble metals** are metals that are resistant to corrosion and oxidation in moist air (unlike most base metal). The short list of chemically noble metals (those elements upon which almost all chemists agree) comprises ruthenium (Ru), rhodium (Rh), palladium (Pd), silver (Ag), osmium (Os), iridium (Ir), platinum (Pt), and gold (Au).

%%%===

2). Since no chemically doped pn junction is required for light-emitting field-effect transistors (LE-FET)" confusing, although today's commercial OLED are often chemically doped, there have been many which are not."

%%%---

The comment is also not grammatically correctly written and difficult to understand what the viewer indeed meant, but we try to answer to the comments. The chemical doping is not made for enhancing carrier injection but instead this is used for enhancing channel transport current by lowering channel resistance. In a doped pn junction, greatly asymmetric carrier injection between holes and electrons is still reported. A good review can be found in (Doping organic semiconductors, S. Lussem et al., Phys. Status Solidi, A210, 9-43 (2013), where a problem of asymmetric and unbalanced carrier injection is explained in the case of chemical doping. For light-emitting FETs, chemical doping is generally employed for not getting better carrier injection, but instead this is used in order to enhance emission efficiency by exciton/energy transfer from a host OSC to a guest OSC (dopant), and therefore the purpose of chemical doping is completely different and ambipolar carrier injection is not improved by chemical doping.

%%%===

#3. In particular, an experimental proof for the main claim and key underlying mechanism, the continuous disordered states (CDS), e.g. by spectroscopy, is not given."

%%%---

Please keep in mind that this comment was not a main concern in the first round of the reviewing process, and we were not asked to reply to this question carefully. We have given experimental evidence of the Bardeen limit conversion as well as CDS in the revised manuscript (carefully see, Supplemental Information S5 and the replies to the comments of Referee #1).

We can consider two types of states inside the gap, and each of them makes very important roles in a different manner in the new electrodes of M/pc-OSC/TTC and can offer the higher ambipolar carrier injection than any other electrodes so far reported. One is the gap states induced by the surface disorder of semiconductors (disorder induced gap states, DIGS) or the metal-semiconductor interface (metal-semiconductor interface induced gap states, MIGS), and the other is the continuously spreading disordered states created by the disorder of molecules inside the gap of bulk polycrystal semiconductors (CDS). In the former, the high density of states inside the gap plays a very important role for pinning, giving the Bardeen limit in terms of the vacuum level shift at a semiconductor surface (*J. Bardeen, Surface States and Rectification at a Metal Semi-Conductor Contact, Physical Review* **71** 717 (1947)). As for the conversion from the Schottky to the Bardeen limit, we clearly showed the important experimental evidence by photoelectron spectroscopy that a large vacuum level shift can occur at the metal-semiconductor (M-S) interface, when the surface texture of OSC is controlled by TTC, which is unambiguous and clear evidence being consistent for the pinned states associated with DIGS/MIGS. It is noted that when the surface of OSC is very smooth and flat, a band bending without any significant vacuum shift reflecting the Schottky limit of MS contact was experimentally observed (T. Kanagasekaran et al., *Appl. Phys. Lett.* **107**, 043304 (2015)).

In order to provide experimental evidence on CDS generated inside the gap of pc-OSC, we have made key experiments as described in the revised manuscript as explained in the 2nd round reviewing process. We compared two different pc-OSCs from the active sc-OCS for the new electrode: Au/pc-RU/TTC and Au/pc-BP2T/TTC constructed on sc-BP2T and Au/pc-RU/TTC and Au/pc-BP2T/TTC constructed on sc-RU. Intriguingly, greatly better ambipolar carrier injection can be observed for Au/pc-RU/TTC on sc-RU and Au/pc-BP2T/TTC on sc-PB2T than that of Au on sc-RU and sc-BP2T, while much worse carrier injection was observed for Au/pc-BP2T/TTC on sc-RU and Au/pc-RU/TTC on sc-BP2T. These experimental data unambiguously show that the energy level matching between the CDS of pc-OSC and VB and CB of sc-OSC is crucial, indicating the importance of CDS made by the molecular disorder. The carriers injected to pc-OSCs via CDS can be released to other states via a small excitation energy, leading to good electrical transport and efficient ambipolar carrier injection (multiple trap and release: MTR, M. Shur and M. Hack., *Journal of Applied Physics*, **55**, 3831–3842, (1984)). According to these new experimental facts, Reviewer #1 has agreed and thinks that this work reserves the publication in

Nature Communications. I believe that the reviewer #4 can understand the important scientific message of our work, as the other reviewers do.

%%%------

Important note to the editorial board:

%%%===

Although Reviewer #2 did not send any report on our revised manuscript even though he/she was in the positive position for recommending publication of this manuscript in Nature Communications, we leave here the replies to the comments again as below. We hope that Reviewer #2 can reply this time for supporting our manuscript for Nature Communications, with keeping his original positive judgment. Otherwise, such an attitude is scientifically very unfair. We wish the editorial board to contact again to the reviewer #2, so that he/she could give fair scientific comments with keeping his/her original statements.

%%%=====

%%%===

[Replies to the comments by the 2nd reviewer]

%%%=====

Reviewer #2 (Remarks to the Author):

This work presents a new concept for highly efficient carrier injection into organic semiconductors using two buffer layers of TTC and pc-OSC. Although the insertion of buffer layer is not new idea, the carrier injection efficiency of this new type electrode is almost equivalent both for hole and electron, which is impossible for previous types of electrodes. The results are very clear and the effectiveness of new electrode is obvious. The main weakness of the present work lies in that the detail measurement to prove the mechanism is seriously lacking. After answering this comments, I will agree with publication of this manuscript in Nature Commun.

%%%===

Thank you very much for taking your valuable time for the reviewing process of our manuscript. We are very pleased to hear your positive comments for agreeing with publication in Nature Communications, after replying your comments. We have replied to the comments and revised our manuscript by referring to the comments as follows:

%%%===

1. Author should clarify the induced IGS by pc-OSC layer using experimental way,

because this is most important point of this research. Although authors explains the possible mechanism, most part is not supported by experimental results.

%%%=

Reply to the 1st comments:

The difference in carrier injection between the Schottky and the Bardeen limit can experimentally be recognized by the fact that the Schottky barrier is strongly dependent on the work function of metal electrodes (ϕ_M), while it is nearly independent of ϕ_M in the case of the Bardeen limit. As for the possible conversion from the Schottky to the Bardeen limit, we previously showed such experimental evidence by photo electron spectroscopy (T. Kanagasekaran et al., *Appl. Phys. Lett.* **107**, 043304 (2015)). When less disordered polycrystal BP2T (pc-BP2T) is grown on a PMMA/SiO₂ substrate, a typical M-S Schottky limit can be observed, while the regulated disorder created by employing a TTC/SiO₂ substrate can modify it to the Bardeen limit. The vacuum level shift at the metal-semiconductor interface was negligible in the former case and consequently the barrier height against carrier injection is determined by the difference between ϕ_M and the conduction band minimum and the valence band maximum for electron and hole injection, respectively. On the other hand, a considerable vacuum level shift was observed in the latter case, indicating a significant deviation from the Schottky limit towards the Bardeen limit.

In order to provide additional experimental evidence for the continuous disordered states (CDS) generated inside the gap of pc-OSC, we have made additional experiments (see, SI: Fig.S5). By searching the same optimum condition of experiments, we compared two different pc-OSCs from the active sc-OCS in the new electrode structure: Au/pc-RU/TTC and Au/pc-BP2T/TTC on sc-BP2T and Au/pc-RU/TTC and Au/pc-BP2T/TTC on sc-RU. Intriguingly, greatly better ambipolar carrier injection can be observed for

Au/pc-RU/TTC on sc-RU and Au/pc-BP2T/TTC on sc-PB2T than that of pure Au on sc-RU and sc-BP2T, while much worse carrier injection was observed for Au/pc-BP2T/TTC on sc-RU and Au/pc-RU/TTC on sc-BP2T. ~~These~~ This experimental data unambiguously gives an important fact that the CDS created by the molecular

Figure III | Band diagram of Au and BP2T in eV without contact (left), and a Au-BP2T interface on PMMA-SiO₂ (center) and TTC-SiO₂ (right). E_F and Δ denote the Fermi level and the vacuum level shift at the interface, respectively. The lines inside the red zone in color indicate the gap states.

disorder in bulk pc-OSCs plays an important role for reducing the barrier height for carrier injection to sc-OSC, where the energy level matching between the CDS created in pc-OSC and VB/CB of sc-OSC is crucial.

Figure S5 | Transfer characteristics of organic single-crystal field-effect transistors with and without interlayer. (a) Rubrene single crystal with Au electrodes (yellow circle), with Au/rubrene/TTC electrodes (magenta square), and Au/BP2T/TTC electrodes (magenta square). (b) BP2T single crystal with Au electrodes (yellow circle), with Au/BP2T/TTC electrodes (magenta square), and Au/rubrene/TTC electrodes (magenta square). (c) BP3T single crystal with Au electrodes (yellow circle), with Au/BP2T/TTC electrodes (violet square).

These two experiments clearly show that the change in the carrier injection limit from the Schottky to the Bardeen due to the high density pinning states and the continuous disordered levels (CDS) created in the bulk pc-OSC are the most important factors for providing the extremely high ambipolar carrier injection. These are unambiguous and sufficient experimental evidence for supporting the mechanism of highly efficient ambipolar carrier injection. We would like to stress on that the mechanism of ambipolar carrier injection of the new electrode is surely of interest, and furthermore it is also greatly important that the highest ambipolar carrier injection among any other electrodes so far reported has been realized in the new concept of electrodes with almost equivalent efficiency between holes and electrons.

%%%===

2. Author should show more experimental results, such as out-put characteristics, the mobility difference between two-terminal and four-terminal measurements, and transistor characteristics before and after air exposure. Particularly, if author want to claim air stability, comparison before and after air exposure is key results.

%%%===

Reply to the 2nd comments:

According to the comments of the reviewer, we have prepared output characteristics,

the mobility difference between two-terminal and four-terminal measurements, and the transistor characteristics before and after air exposure as Supplemental Information. Just in case for the reviewer, we also list the same data here.

Figure S1 | Four-terminal (cyan circles) and two-terminal (red squares) mobilities of single-crystalline organic semiconductor (sc-OSC) field-effect transistors with M/CsF, M, M/polycrystalline-OSC (pc-OSC) and M/pc-OSC/ tetratetracontane (TTC) electrode (M = Au or Ca). (a, b) P-type and n-type mobilities of rubrene. (c, d) P-type and n-type mobilities of 5,5'-di(4-biphenyl)-2,2'-bithiophene (BP2T). Reliable values were not obtained for some transistors because of large contact

Figure S2 | Output characteristics of single-crystalline Rubrene (sc-RU) field-effect transistors with M, M/pc-RU and M/pc-RU/tetratetracontane (TTC) electrode (M = Au or Ca). (a) P-type characteristics. (b) N-type characteristics. Left panels: M electrode. Center panels: M/pc-RU electrode. Right panels: M/pc-RU/TTC electrode. Top panels: M = Au. Bottom panels: M = Ca.

Figure S3 | Output characteristics of single-crystalline 5,5'-di(4-biphenyl)-2,2'-bithiophene (sc-BP2T) field-effect transistors with M, M/pc-BP2T and M/pc-BP2T/tetratetracontane (TTC) electrode (M = Au or Ca). (a) P-type characteristics. (b) N-type characteristics. Left panels: M electrode. Center panels: M/pc-BP2T electrode. Right panels: M/pc-BP2T/TTC electrode. Top panels: M = Au. Bottom panels: M = Ca.

Figure S5 | Transfer characteristics of organic single-crystal field-effect transistors before (black) and after (red) air exposure of 2 h. (a) Rubrene single crystal (sc-RU) with Au/polycrystalline rubrene/TTC electrodes. (b) BP2T single crystal (sc-BP2T) with Au/polycrystalline BP2T/TTC electrodes.

%%%===

3. The evaluation of injection efficiency is not quantitative. The temperature dependence of IV characteristic is one possible method to evaluate injection efficiency.

%%%---

Reply to the 3rd comments:

In order to evaluate the injection efficiency, we have carried out measurements on temperature dependences of contact resistances. From the experimental results, we obtained carrier injection barriers and summarized in Fig. S4. It is clearly shown that the pc-OSC layer decreases the injection barrier of both hole and electron, and the insertion of the TTC layer further decreases the injection barrier. The carrier injection barriers of M/pc-OSC/TTC of unfavorable metal M are comparable to those of favorable metal without pc-OSC/TTC layer.

Figure S4 | Activation energies of carrier injection to single-crystalline organic semiconductor (sc-OSC) field-effect transistors with M, M/pc-OSC and M/pc-OSC/ tetratetracontane (TTC) electrode (M = Au or Ca). (a, b) Hole and electron injection into a rubrene single crystal. (c, d) Hole and electron injection into a rubrene single crystal. Activation energies of Ca electrodes for hole injection and a Au electrode for electron injection into BP2T could not be obtained because of too large contact resistances.

%%%===

4. I think the arguments on two terminal mobility (comparisons and the statement of high mobility) is not important because the main point of this research is injection efficiency and two terminal mobility is also dependent on crystal quality. Therefore, contact resistance is key parameter. However, the contact resistance shown in Fig. 3c is not small, comparing with the previous champion report. Please explain the reasons.

%%%---

Reply to the 4th comments:

The reported contact resistances, which are lower than that in the present report, are in the case of the bottom-gate bottom-contact configuration, while the FET configuration in the present work is the bottom-gate top-contact. We described that the contact resistance in the present report is the lowest among FETs with bottom-gate top-contact configuration. The difference in contact resistance between the two configurations is attributed to the distance between electrodes and accumulation layers. In the present work, we focused on the improvement in carrier injection at the metal-semiconductor interfaces without changing electrode configurations.

%%%===

5. The effect of induced damage during deposition process is not negligible. The comparison between top and bottom contact is another important approach.

%%%---

Reply to the 5th comments:

Damage created during the metal deposition process sometimes strongly depends on the process. When a condition is well regulated, good injection of even holes from Ca electrodes and electrons from Au electrodes occurs via CDS in spite of the Schottky barriers when the new electrodes are employed. We have also confirmed that no serious influences are observed when we correctly regulate the evaporation conditions. Regardless of the possible damages, we can optimize the process so that the highest carrier injection so far reported was achieved.

As the reviewer considered, we also thought that the comparison with top-gate and bottom-contact configurations will give further detailed information, and have tried to fabricate FETs with the bottom-contact configuration. However, good electrostatic sticking was not obtained in the case of sc-OSCs of RU and BP2T on the electrodes covered by pc-OSC in the new electrode structure as shown in the left panels of Fig. IV, and a good device fabrication process was not achieved as shown in the right panels of Fig.IV. The application of the new electrodes for a device with the bottom-contact

structure depends on the electrostatic conditions of sc-OSCs, and technical improvements have to be considered.

Figure IV | Optical micrographs of sc-OSC on Au electrodes covered with pc-OSC.

%%%=

6. The statement of “bright emission” without quantitative results is not fair. Therefore, if author want to claim “bright emission”, the EQE of fabricated device is necessary. Moreover, I am not sure that it is really ambipolar light emission because channel length is too short. The demonstration with longer channel transistor is also required.

%%%---

Reply to the 6th comments:

The statement of “bright emission” is used just in comparison with single-crystal FETs of the same material with Ca and Au hetero-electrodes. The statement is rationalized by the fact of the large ambipolar current, by supposing that the EQE is comparable as long as a high quality sc-OSC is used as the transporting and the emitting layer.

As for the latter comment, in order to show a fact that the emission of ambipolar carriers occurs in the carrier transporting layer, we replaced the original figure by a new optical micrograph of light emission from FETs with longer channel length in Fig. 4. As clearly seen, emission apart from the electrodes is observed, whose position moves as V_G changes. This shows that the emission is an ambipolar light emission.

Fig.4 Light emission from a FET of a rubrene single crystal (sc-RU) with Au/pc-RU/TTC electrodes. Top panels: an optical micrograph (top view) and schematic illustration of a sc-RU FET. Lower panels: Change in position of light-emission zone and luminescence intensity with respect to the gate voltage (V_G).

%%%*****

%%%*****

Record of the comments on the 2nd revision.

%%%*****

Reviewer #1 (Remarks to the Author):

While I remain skeptical about the mechanism proposed by the authors, I believe the phenomena they report is interesting, potentially important (given the multiple systems in which it operates) and merits further investigation by the broader community. I do wish that they addressed, in the revised manuscript, the possible role that field emission and tunneling play in what is observed (including arguments that in their view rule this out). Nevertheless, their revision does a better job of laying out their proposed mechanism so that it can be more readily understood, and if they are correct, it will be an important result. I therefore lift my previous objection to publication in Nature Communications. I do note one item that they must fix in their supporting information Figure S3. They label the horizontal axis of their output curves as "Gate Voltage". This should be Drain Voltage.

%%%=====

Reviewer #3 (Remarks to the Author):

After carefully reading the manuscript as well as the response to the comments of previous two reviewers, I believe that the ms is potentially interesting to the target

audience of Nature Communications. Especially, the authors have followed the constructive comments of Reviewer 2 to further explain the mechanisms.

However, I still have one concern: How general is this new electrode approach ~~is~~?

Does it only work for some specific cases (e.g. BP2T6 and rubrene as shown by the authors)? Or is it a general approach which works for OFETs? For the later case, I would suggest to accept the ms. Otherwise, I would think that it fits better to a specialized journal.

###===

Reviewer #4 (Remarks to the Author):

The paper „New concept of electrode in organic semiconductors“ by Thangavel Kanagasekaran et al reports a novel approach to contact organic single crystals with a polycrystalline layer, SAM, and metal electrode. The data clearly show that these contacts work very well and are a significant advance compared to previous approaches, although one of the claims, that ignoble metals can be used, is not really fulfilled since Ca is still present. Despite the excellent experimental work on many different transistor structures, the underlying physics is limited. In particular, an experimental proof for the main claim und key underlying mechanism, the continuous disordered states (CDS), e.g. by spectroscopy, is not given. I thus believe that the paper does not really has the level needed for Nature Communications. Maybe „Scientific Reports“ would fit. Furthermore, I would add two issues before the paper is published:

- A very thorough language polish is needed: As example, I cite the following sentence „This has been one of major problems for a long year that injection of electrons is greatly difficult compared to those of holes in OSCs since stable metals with low E F , such as Au, Cu, Ag, are typically employed.“

- I find the sentence p.4 l.50 „Such ambipolar injection and subsequent light emission have attracted much attention from the viewpoint of optoelectronics since no chemically doped pn junction is required for light-emitting field-effect transistors (LE-FET)“ confusing, although todays commercial OLED are often chemically doped, there have been many which are not.

REVIEWERS' COMMENTS:

Reviewer #3 (Remarks to the Author):

I would suggest to accept the ms.

Reviewer #4 (Remarks to the Author):

I thank the authors for clarifying the point about Ca, this was indeed a misunderstanding from my side. My other point concerning the proof for the new physics claimed is unfortunately not rebutted so that I stand by my earlier judgment about the paper.